# Application of machine learning in *in vitro* propagation of endemic *Lilium akkusianum* R. Gämperle

**Mehmet Tütüncü** [ORCID] *

Department of Horticulture, Faculty of Agriculture, University of Ondokuz Mayıs, Samsun, Türkiye

* mehmet.tutuncu@omu.edu.tr

**Data Availability Statement:** All relevant data are within the manuscript and its Supporting Information files.

## Abstract

A successful regeneration protocol was developed for micropropagation of *Lilium akkusianum* R. Gämperle, an endemic species of Türkiye, from scale explants. The study also aimed to evaluate the effects of Meta-Topolin (mT) and N6-Benzyladenine (BA) on *in vitro* regeneration. The Murashige and Skoog medium (MS) supplemented with different levels of α-naphthaleneacetic acid (NAA)/BA and NAA/mT were used for culture initiation in the darkness. The highest callus rates were observed on explants cultured on MS medium with 2.0 mg/L NAA + 0.5 mg/L mT (83.31%), and the highest adventitious bud number per explant was 4.98 in MS medium with 0.5 mg/L NAA + 1.5 mg/L mT. Adventitious buds were excised and cultured in 16/8 h photoperiod conditions. The highest average shoot number per explant was 4.0 in MS medium with 2.0 mg/L mT + 1.0 mg/L NAA. Shoots were rooted with the highest rate (90%) in the medium with the 1.0 mg/L IBA, and the highest survival rate (87.5%) was recorded in rooted shoots in the same medium. The ISSR marker system showed that regenerated plantlets were genetically stable. Besides traditional tissue culture techniques used in the current study, the potential for improving the effectiveness of *L. akkusianum* propagation protocols by incorporating machine learning methodologies was evaluated. ML techniques enhance lily micropropagation by analyzing complex biological processes, merging with traditional methods. This collaborative approach validates current protocols, allowing ongoing improvements. Embracing machine learning in endemic *L. akkusianum* studies contributes to sustainable plant propagation, promoting conservation and responsible genetic resource utilization in agriculture.

## Introduction

The members of the genus *Lilium* are perennial, herbaceous, and bulbous plants belonging to the Liliaceae family. More than 100 species represent the genus, primarily spread in temperate regions in the northern hemisphere [1]. It is well known that lilies have been used as ornamental plants in European gardens for centuries and decorative purposes since ancient times in Egypt and the Middle East [2]. *Lilium* species are still economically valuable ornamental plants with attractive and large flowers, which play an essential role in the cut flower industry.

**Funding:** The author(s) received no specific funding for this work.

**Competing interests:** The authors have declared that no competing interests exist.

In Türkiye, the genus *Lilium* is represented by seven species (*L. akkusianum* R. Gämperle (endemic), *L. candidum* L., *L. ciliatum* P. H. Davis (endemic), *L. kesselringianum* Miscz., *L. martagon* L., *L. monadelphum* M. Bieb., and *L. ponticum* K. Koch.). *L. akkusianum* is one of the endemic *Lilium* species discovered by Ernst Gügel and Heinz-Werner Zaiss in 1993 and introduced to the scientific world by René Gämperle in 1998. It is distributed in a restricted area of the Akkuş district of Ordu province and on the border of Tokat province within Türkiye [3]. *L. akkusianum* genotypes are naturally grown in moist deciduous forests at about 1200–1400 m. The species is often confused with *L. candidum* but is easily distinguished by some morphological features. The bulbs of *L. akkusianum* are similar to those of *L. candidum*, but they are whiter, and their scales are more lanceolate. The plant height of the genotypes is up to 1 m, the stem diameter is about 8–18 mm, and the number of flowers is between 1–14. The cream flowers are oval-elliptical and slightly curved back. Leaf margins and flower buds are woolly hairy. The plant blossoms in June and July, and their flower life is longer [4]. Turkish lilies are mainly grown in the northern part of the country and are threatened by their restricted geographical ranges and small population sizes [5]. Additionally, *L. akkusianum* and other *Lilium* species are at risk of extinction due to urbanization in highlands and the destruction of nature, overgrazing, unconscious over-harvesting of bulbs from nature and forest fires [6]. Therefore, urgent protection measures are needed for this valuable endemic plant.

*In vitro* regeneration techniques offer potential solutions for protecting threatened genetic resources [7], leading to speedy large-scale commercial plant production and standardized, disease-free planting material [8]. Additionally, plant breeders and nurseries frequently employ *in vitro* techniques to accelerate the production of new clones with desirable traits. Although many parts of the plant may be used in the *in vitro* propagation of *Lilium*, *in vitro* scale culture is a superior propagation method for lilies [9]. However, *in vitro* regeneration from scale explants relies on many factors, such as cytokinin, auxin, and sucrose concentrations [10, 11]. Among the various plant growth regulators, the most efficient approach for establishing cultures and achieving *in vitro* mass propagation of lilies is carefully using appropriate concentrations of auxins and cytokinins [12]. Prior research findings have suggested that auxins 2,4-D (2,4-Dichlorophenoxyacetic acid), NAA (1-Naphthaleneacetic acid), IAA (Indole-3-acetic acid) and cytokinins Thidiazuron (TDZ) and BA (N6-Benzyladenine) are predominantly used as in the micropropagation of lilies viz., *L. pumilum* [13], Oriental Lilium Hybrid Cv. 'Ravenna' [14], *L. ledebourii* [15], *L. longiflorum* [16], *L. leucanthum* [17], *L. monodelphum* [18], *L. candidum* [19], *L. martagon* [20], and *L. longiflorum* [21].

Properly used cytokinins are crucial in regulating the cell cycle, developing plants and cell proliferation in micropropagation. However, exposing plants to inappropriate types or concentrations of cytokinins can adversely affect mass propagation [22]. In the past two decades, the application of meta-topolin (mT), originally extracted from poplar leaves, gained popularity in *in vitro* plant propagation [23]. It has been indicated that difficulties like hyperhydricity, shoot-tip necrosis, poor rooting ability, and early senescence while creating effective micropropagation protocols can be alleviated using mT alternative to BA [24]. Additionally, positive effects of mT in micropropagation have been proven in shoot regeneration of *Daphne mezereum* [25], *Gerbera jamesonii* [26], *Iris × hollandica* [27], and *Tulipa* sp. L. [28], *Hypericum* sp. L. [29].

The tissue culture process encompasses various elements, such as the composition of the growth medium and environmental conditions, which can introduce genetic variations among regenerated plants. Therefore, it is paramount to ensure the genetic consistency of these regenerated plants. This measure is critical to facilitate the successful proliferation and preservation of rare plant species [30]. In recent years, Polymerase Chain Reaction (PCR)

techniques, specifically Inter-Simple Sequence Repeats (ISSR), have proven to be effective in assessing the genetic fidelity of regenerated plantlets in lilies [31–33].

Machine learning (ML) modeling has become popular in plant biotechnology. However, more data is needed to support this claim. This is especially true for *in vitro* plant tissue culture. Different machine learning algorithms have been used to optimize a variety of aspects, including *in vitro* sterilization [34], *in vitro* germination [35], *in vitro* cell culture [36], and *in vitro* somatic embryogenesis [34].

Like the human brain, artificial neural networks (ANNs) comprise linked processing neurons that may interpret information via weighted communication links [37]. Artificial neural networks (ANNs) are used to empirically establish links between independent (input) and dependent (output) variables, making it easier to analyze the information exchange between the two.

One of the most prominent artificial neural networks (ANNs) is the multilayer perceptron (MLP), which is organized into several levels, including an input layer, an output layer, and one or more hidden layers. Using both input and output variables as part of the training set, MLP uses a supervised training technique, Eq 1.

XGBoost algorithm is a powerful tool for solving problems with classification and regression. It is well known for its exceptional performance and speed and is a member of the Gradient Boosting Decision Trees class. Within a gradient-boosting architecture, XGBoost excels at learning from errors and progressively reducing the error rate over multiple iterations.

Successful shoot regeneration protocols have been reported for widely grown native *Lilium* species, but studies on mT-mediated *in vitro* regeneration in *Lilium* are very limited. Notably, studies have not reported establishing an efficient mass clonal propagation protocol for *L. akkusianum* nor assessed the regenerated plants' genetic stability. Therefore, this study aims to establish the *in vitro* regeneration potential of endemic *L. akkusianum* species from bulb-scale explants, comparing the effects of the commonly used BA with its new alternative mT in adventitious shoot regeneration. Additionally, this study seeks to incorporate ANN analysis and machine learning methodologies to broaden the research scope. Employing computational techniques, the research aims to model and forecast the impacts of diverse culture media components and plant growth regulators on the quality of micropropagation.

## Materials and methods

### Plant material and surface sterilization

*L. akkusianum* bulbs were gathered from the natural habitat in Akkus province, Ordu, Türkiye after the flowering season in 2021 with the official permission numbered 21264211–288.04-E.817744 obtained from the Ministry of Agriculture and Forestry, the Republic of Türkiye. The bulbs were washed to remove soil, and roots were trimmed off from the bulbs. The excessive water was drained, and bulbs were stored at 4°C for two months before *in vitro* culture. After storage, the scales were washed under running tap water for two hours and pre-sterilized with $HgCl_2$ (0.1%) for 20 min. The scales were rewashed with sterile distilled water and transferred to sterile conditions for further sterilizations. The scales were immersed in a 70% ethanol solution for 1–2 minutes and rinsed multiple times with sterile water to remove residual ethanol. They were then soaked in a 25% commercial bleach solution (Domestos®, 4.5% v/v) for 15 minutes, and the final step involved four additional washes with sterile distilled water. Following the surface sterilization process, the outer layers of the scales were carefully removed, leaving only the inner fleshy layer. This inner layer of the scale was then cut into small squares, approximately 0.5 cm x 0.5 cm, using a sterile scalpel. (Fig 1A).

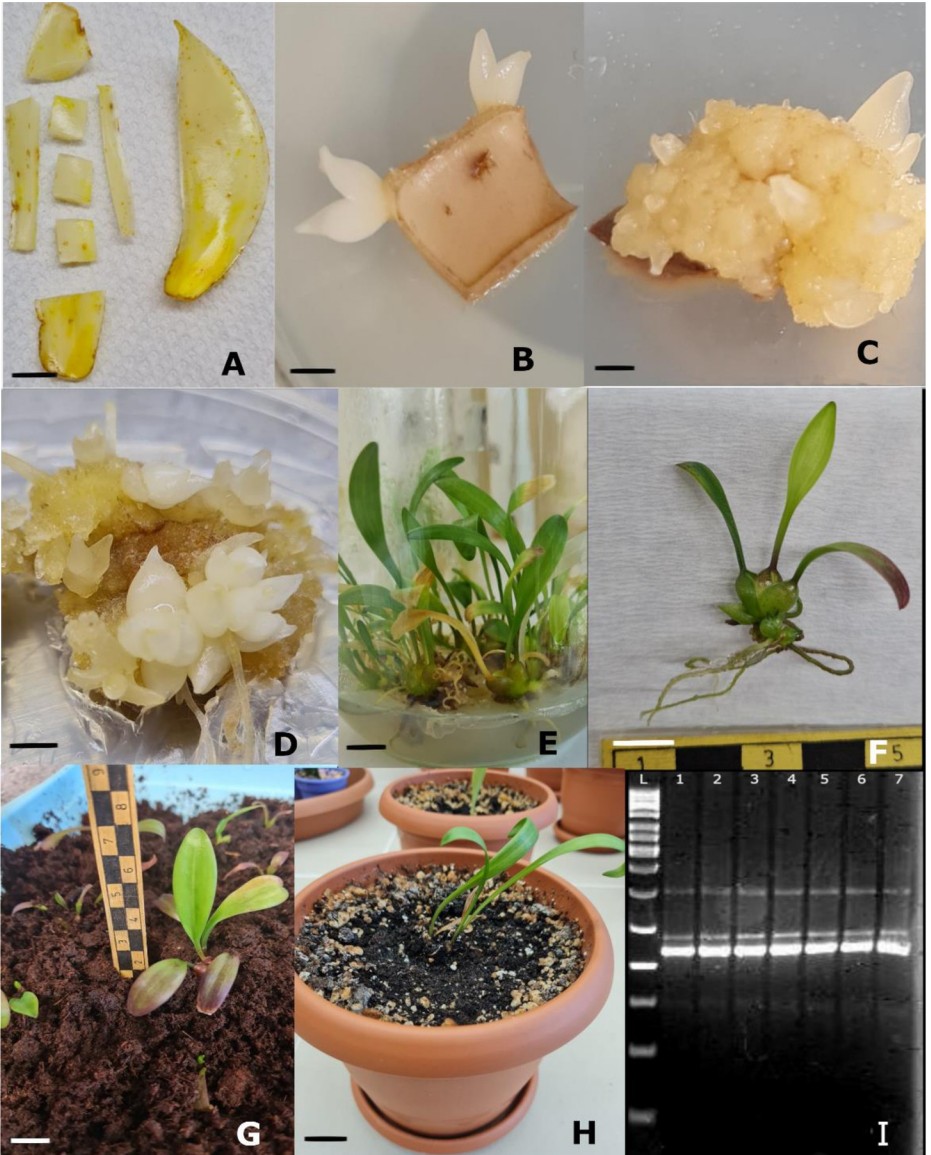

**Fig 1. *In vitro* regeneration in *L. akkusianum*.** (A) Explant preparation. (B) Adventitious bud regeneration in M0 medium 5–6 weeks of culture in the dark. (C) Callus and adventitious buds regeneration in M9 medium 5–6 weeks of culture. (D) Callus and adventitious buds regeneration in M10 medium 5–6 weeks of culture. (E) Shoots development and proliferation under 16/8 h photoperiod (F) Plantlets with well-developed roots. (G) Acclimatized plantlets in the grow box. (H) Plantlets 2 months after transfer into the peat-perlite substrate (1:1 v/v). (I) DNA bands profile of *in vitro* regenerated plantlets (2–7) and mother plant (1). Scale bars: B, C, D, E = 10 mm; A, F, G = 1 cm; H = 5 cm.

## Culture initiation and growth condition

The explants were cultured in Petri dishes (90 × 15 mm) containing 25 ml of full-strength MS medium [38] supplemented with 3% sucrose containing different concentrations and a combination of α-naphthaleneacetic acid (NAA) (0.5, 1.0, 2.0 mg/L) with 6-benzyl amino purine (BA) (0.5, 1.0, 2.0 mg/L) or Meta-Topolin (mT) (0.5, 1.0, 2.0 mg/L). Twelve different culture media were tested, each with seven replicates (Petri dishes) and each replicate containing four explants. The medium was solidified using 7 g/L agar, and the pH of the medium was adjusted

to 5.6–5.8 before autoclaving at 121˚C and 15 psi for 15 min. All explants were cultured at 23± 2˚C in constant darkness for 2 months and subcultured every 4 weeks.

After the first subculture, the explants started producing adventitious bud or callus. They were transferred to a fresh culture medium for further growth and development. After regeneration induction from scales, callus rate (%) and adventitious bud per explant were recorded.

## Effects of different mT and BA on shoot proliferation

Adventitious buds were excised and cultured in the jar (0.6 L) (Fig 1E) containing MS medium containing 1 mg/L NAA + 2 mg/L mT, 1 mg/L NAA + 2 mg/L BA and 0.5 mg/L NAA + 1 mg/L mT which exhibited the highest number of adventitious buds per explant during initiation stage. Each culture medium has nine replicates (jars) and each replicate containing four explants. The cultures were maintained in a growth room at a temperature of 23± 2˚C, with a light/dark cycle of 16 hours of light and 8 hours of darkness. Cool white fluorescent lamps provided illumination with a photosynthetic photon flux density (PPFD) of 70 μmol/m$^2$s. The explants were cultured for 3 months in the same medium, and shoot length (cm), shoot fresh weight (g) and shoot number per explant were observed.

## *In vitro* rooting and acclimatization

Mature shoots of *L. akkusianum* were carefully cut from the clusters of shoots and then transplanted into jars (0.6 L) containing 50 ml MS medium with varying concentrations of NAA or IBA (0.5, 1.0, 2.0, 3.0 mg/L) to induce root development. Seven different rooting media with five replicates, each replicate having four explants, were used in total. All cultures were kept in the growth room under the same conditions with a shoot proliferation stage for six weeks. Then, rooting rate, root number per explant, and root length were measured.

Plantlets were picked up from the culture medium, and residues of the culture medium were removed by washing them under running tap water (Fig 1F). After removing residues, plants were dipped in fungicide solution (Captan 50WP) at 2.5 g/L for 10–15 s and transferred to a grow box with a plastic cover containing sterile peat-perlite substrate (1:1 v/v). The plants were kept in the grow box (Fig 1G) in the growth room at 23± 2˚C under a 16 h photoperiod and 70 μmol/m$^2$s cool white fluorescent lamps for 2 weeks and then transferred to greenhouse conditions, and the plastic cover was gradually removed. After adaptation to greenhouse conditions, the plantlets were transferred to plastic pots (0.9 L) (Fig 1H) containing peat-perlite substrate (1:1 v/v) and irrigated twice weekly with 200 ml water per pot. No chemicals were used during cultivation for pests and diseases.

## Genetic stability of *in vitro*-grown plantlets

The genetic stability of the *in vitro*-grown plantlets was evaluated using the ISSR marker system. Leaves from six *in vitro*-grown plantlets and a mother plant (seven genotypes) were collected and frozen in liquid nitrogen. DNA isolation was performed based on a protocol for minipreps described by Edwards et al. [39]. The quality and quantity of genomic DNA were evaluated spectrophotometrically (NanoDrop ND 100, NanoDrop Technologies). Fourteen ISSR primers (Table 1) were used to assess genetic stability [40]. Amplification was carried out in a volume of 25 μL (10 ng of L. akkusianum DNA template, 2X PCR Mastermix (Fermentas K0171, Waltham, MA, USA), 25 mM MgCl$_2$, 10 μM primer, and 1 U of Taq DNA polymerase (Fermentas EP0402)) according to following program in a Veriti thermal cycler (Applied Biosystems): initial denaturation at 94˚C/3 min, 35 cycles of 94˚C/45 sec, 55˚C/1 min, 72˚C/45 sec, and final elongation at 72˚C/7 min. PCR products were then held at 4˚C.

**Table 1. ISSR primer sequence [40].**

| No | Primer code | Sequence (5'-3') |
|----|-------------|------------------|
| 1 | UBC807 | AGA GAG AGA GAG AGA GT |
| 2 | UBC808 | AGA GAG AGA GAG AGA GC |
| 3 | UBC810 | GAG AGA GAG AGA GAG AT |
| 4 | UBC811 | GAG AGA GAG AGA GAG AC |
| 5 | UBC812 | GAG AGA GAG AGA GAG AC |
| 6 | UBC815 | CTC TCT CTC TCT CTC TG |
| 7 | UBC816 | CAC ACA CAC ACA CAC AT |
| 8 | UBC818 | CAC ACA CAC ACA CAC AG |
| 9 | UBC820 | GTG TGT GTG TGT GTG TC |
| 10 | UBC827 | ACA CAC ACA CAC ACA CG |
| 11 | UBC834 | AGA GAG AGA GAG AGA GT |
| 12 | UBC835 | AGA GAG AGA GAG AGA GC |
| 13 | UBC845 | CTC TCT CTC TCT CTC TG |
| 14 | UBC850 | GTG TGT GTG TGT GTG TC |

The amplified products were separated on 1.5% agarose gel in 0.5X TAE buffer (40 mM Tris-acetate, 1 mM EDTA, pH 8.0) for 3.5 h at 80 V. The gels were stained with 0.5 g/mL ethidium bromide. A 1 kb DNA ladder (Fermentas) was used to determine the fragment size. Fragment patterns were photographed by a gel documentation system (Genesys) under UV light. DNA band profiles resulting in ISSR analysis were visually checked. It was determined whether there were any differences between the control sample and those obtained from tissue culture studies.

## Experimental design and statistical analysis

Experiments were established based on a completely randomized design. Means of all data were separated by variance analysis, and significant differences ($p < 0.05$) were evaluated with an LSD test using the JMP® program (SAS Institute, Cary, NC). The percentage values were arc-sin transformed before the variance analysis.

## XGBoost and MLP analysis

Extreme Gradient Boosting (XGBoost) and Multilayer Perceptron (MLP) were used in this study to model and predict adventitious bud induction, shoot proliferation, and rooting from scale explants of *L. akkusianum*. The dataset was split into training and testing subsets using a five-fold cross-validation technique to evaluate the MLP and XGBoost's predictive performance thoroughly.

The target (output) variables included adventitious buds, shoot length, number of shoots, shot weight, root length number of rooting plants, and number of roots. The input variables included rooting and shooting media. Caret and Kernlab packages were utilized to facilitate the coding process using the R programming language.

Metrics such as the coefficient of determination ($R^2$), which calculates the strength of the relationship between the model and dependent variable, Root Mean Square Error (*RMSE*), which indicates how closely the regression line matches the observed data points, and Mean Absolute Error (*MAE*), which calculates the average amount of error between the observed and predicted values, were used to assess and compare the efficacy and precision of the models.

Eqs 1–3.

$$R^2 = 1 - \frac{\sum_{i=1}^{n} (Y_i - \hat{Y}_i)^2}{\sum_{i=1}^{n} (Y_i - \tilde{Y})^2} \tag{1}$$

$$RMSE = \sqrt{\frac{(\sum_{i=1}^{n} (Y_i - \hat{Y}_i)^2)}{n}} \tag{2}$$

$$MAE = \frac{1}{n} \sum_{i=1}^{n} |Y_i - \hat{Y}_i| \tag{3}$$

While $Y_i$ = actual value, $\hat{Y}_i$ = Predicted value, $\tilde{Y}$ = mean o the actual values and $n$ = sample count.

## Results

### Regeneration capacity of scale explants

The scale explants of *L. akkusianum* were cultured on the MS medium containing auxin and cytokinin combinations to induce *in vitro* regeneration (Fig 1A). In the fourth week of the culture, explants began to swell, and yellowish and friable callus tissues formed after the four to six weeks of the culture. At this stage, adventitious buds developed directly from explants or callus structures (Fig 1B–1D). Variance analysis of callus induction rate (%) and adventitious bud number per explant has shown that the effects of plant growth regulators (PGRs) were statistically significant (Table 2). Compared with control treatments (MS medium without PGRs), culture medium supplemented with PGRs induced callus formation.

The highest callus rates were observed on explants cultured on MS medium supplemented with 2.0 mg/L NAA + 0.5 mg/L mT (83.31%) and MS medium with 2.0 mg/L NAA + 0.5 mg/L BA (73.8%) while callus induction rate in the MS medium without PGRs (M0) was the lowest. After six weeks of culture initiation, adventitious bud regeneration was observed in all culture

**Table 2. Callus induction rate and adventitious bud number per explants cultured on the MS medium supplemented with various concentrations of NAA and BA or mT.**

| Medium No | NAA (mg/L) | BA (mg/L) | mT (mg/L) | Callus (%) ± SE* | Adventitious bud/explant ± SE ** |
|-----------|-----------|-----------|-----------|------------------|----------------------------------|
| M0 | - | - | - | 0.0 ± 0.0 g (3.55) | 0.14 ± 0.07 g |
| M1 | 0.5 | 0.5 | - | 39.48 ± 9.1 b-e (38.43) | 1.04 ± 0.19 ef |
| M2 | 1.0 | 0.5 | - | 46.4 ± 7.1 bcd (42.74) | 0.59 ± 0.07 fg |
| M3 | 2.0 | 0.5 | - | 73.8 ± 7.2 a (63.52) | 0.54 ± 0.06 fg |
| M4 | 0.5 | 1.0 | - | 14.25 ± 5.2 fg (16.93) | 1.49 ±0.22 cde |
| M5 | 1.0 | 1.0 | - | 26.17 ± 10.5 def (24.51) | 1.21 ±0.20 de |
| M6 | 1.0 | 2.0 | - | 47.6 ± 7.6 bc (43.50) | 2.61 ±0.19 b |
| M7 | 0.5 | - | 0.5 | 29.97 ± 10.9 c-f (28.56) | 1.08 ±0.15 ef |
| M8 | 1.0 | - | 0.5 | 49.97 ± 4.7 bc (44.98) | 1.64 ± 0.14 cd |
| M9 | 2.0 | - | 0.5 | 83.31 ± 4.4 a (69.49) | 1.64 ± 0.14 cd |
| M10 | 0.5 | - | 1.0 | 9.48 ± 3.1 fg (13.73) | 4.98 ± 0.33 a |
| M11 | 1.0 | - | 1.0 | 20.78 ± 3.0 ef (26.76) | 1.87 ± 0.11 c |
| M12 | 1.0 | | 2.0 | 51.17 ± 5.4 b (45.74) | 2.97 ± 0.20 a |

*LSD $_{Callus}$: 6.23, LSD $_{Adventitious\ bud}$: 0.414, Levels not connected by the same letter are significantly different ($p < 0.05$), SE: Standard error.

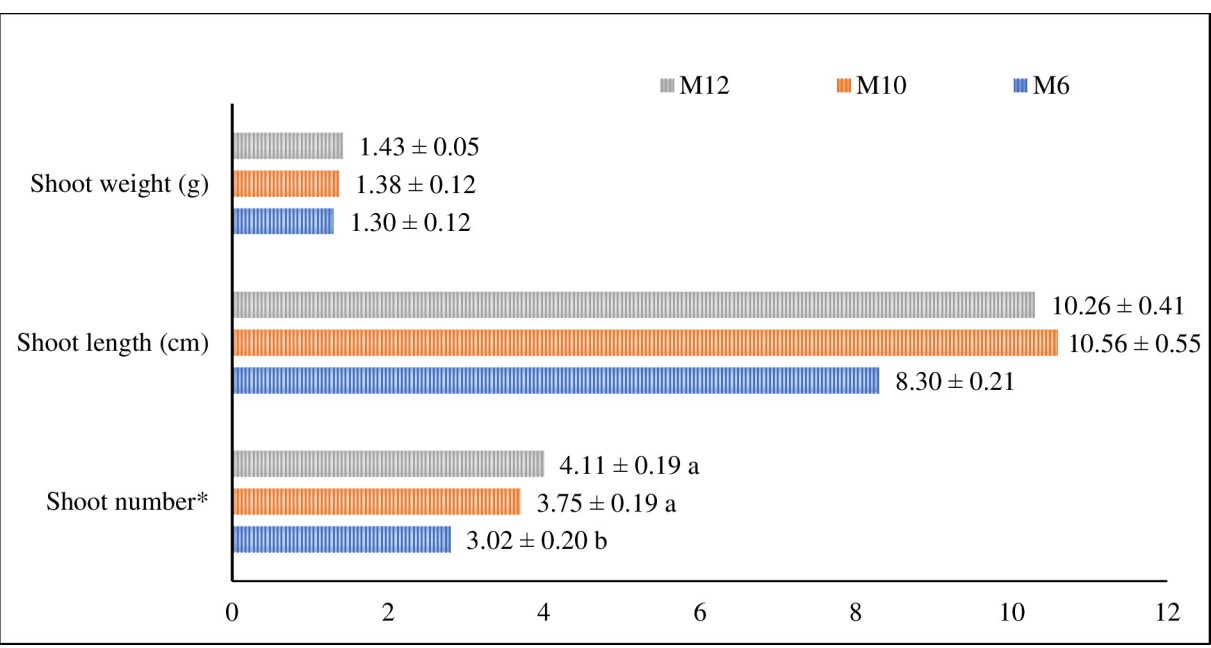

**Fig 2. Effects of plant growth regulator combination on shoot number, fresh shoot length and shoot weight in the sixth week of culture** (*$p < 0.05$, LSD Shoot Number: 0.876; Levels not connected by the same letter are significantly different).

mediums. The addition of PGRs to the medium increased the average number of adventitious bud formation per explant. The highest adventitious bud number per explant was 4.98 in M10 treatments (MS + 0.5 mg/L NAA + 1.5 mg/L mT), and the lowest was 0.14 in M0 (control).

## The effects of different cytokinin on shoot proliferation

Adventitious buds were excised and cultured on the same medium individually at 23± 2°C under 16/8 h light/dark photoperiod to assess their effects on shoot development. Culture mediums had no statistically significant effects of on shoot weight and length. However, the impact of culture mediums on the average shoot number per explant was statistically significant (Fig 2). Average fresh shoot weight varied from 1.3 g to 1.43 g in MS medium supplemented with 2.0 mg/L NAA + 1.0 mg/L BA (M6) and 2.0 mg/L NAA + 1.0 mg/L mT (M12), respectively. The highest average shoot length was 10.56 cm observed in the MS medium supplemented with 0.5 mg/L NAA + 1.0 mg/L mT, while the lowest was in the M6 medium with 8.3 cm. The effects of treatments on shoot number were statistically significant. The highest average shoot number per explant ranged from 3.02 to 4.11 in M10 and M12 medium, showing that MS medium supplemented with mT instead of BA was a superior culture medium for shoot proliferation.

## *In vitro* rooting and acclimatization of plantlets

Effects of NAA and IBA on average rooting percentage, root number per explant, root length and survival rate of the plantlets were statistically significant (Table 3). Compared with the control group, adding auxin into the medium increased the rooting capacity of the explants. The shoots cultured on MS medium supplemented with 1.0 mg/L and 2.0 mg/L NAA or IBA had a superior rooting rate, root number, root length and survival rate than other treatments. However, the highest rooting percentage (90%), the average root number per explant (2.8),

**Table 3. Effects of different NAA and IBA on rooting behavior and survival rate of plantlets.**

| Rooting Medium | NAA (mg/L) | IBA (mg/L) | Rooting (%)* | Root number per explant** | Root length per explant*** | Survival rate (%)**** |
|---|---|---|---|---|---|---|
| R1 | 0.5 | - | 50.0 ± 11.3 cd (45.0) | 0.95 ± 0.23 cd | 1.25 ± 0.28 cd | 12.5 c (11.25) |
| R2 | 1.0 | - | 70.0 ± 10.2 abc (63.0) | 1.55 ± 0.25 bc | 2.13 ± 0.31 bc | 37.5 bc (33.75) |
| R3 | 2.0 | - | 75.0 ± 9.6 abc (67.5) | 1.6 ± 0.24 bc | 2.37 ± 0.32 b | 25.0 bc (22.50) |
| R4 | - | 0.5 | 60.0 ± 10.9 bc (54.0) | 1.15 ± 0.23 cd | 1.66 ± 0.31 bcd | 37.5 bc (33.75) |
| R5 | - | 1.0 | 90.0 ± 6.7 a (81.0) | 2.8 ± 0.32 a | 3.83 ± 0.33 a | 87.5 a (78.75) |
| R6 | - | 2.0 | 80.0 ± 8.9 ab (72.0) | 2.25 ± 0.35 ab | 3.82 ± 0.33 a | 62.5 ab (56.25) |
| R7 | - | - | 30.0 ± 10.2 d (27.0) | 0.4 ± 0.14 d | 0.78 ± 0.27 d | - |

Levels not connected by the same letter are significantly different ($p < 0.05$).

*LSD $_{Rooting}$: 23.39

**LSD $_{Root\ number}$: 0.70

***LSD $_{Root\ length}$: 0.83

****LSD $_{Survival\ rate}$: 26.16.

and the average root length per explant (3.83 cm) were recorded in the treatment of MS medium supplemented with 1.0 mg/L IBA (R5 medium). The lowest values for the parameters were observed in the R7 medium (MS medium without PGRs). Rooted plantlets were hardened in a grow box containing peat-perlite substrate (1:1 v/v), and the survival rates were recorded two weeks later. The highest survival rate was 87.5% and 62.5% in R5 and R6 media, respectively. No plantlets survived in the control group two weeks after hardening. Overall, the results of the rooting experiment showed that IBA treatments are better than NAA.

## Genetic stability of the regenerated plantlets

The genetic stability of *in vitro* adventitious bud-derived plantlets was tested using ISSR markers. According to agarose gel electrophoresis results, DNA profiles of regenerated plantlets and mother plants did not differ. An ISSR gel image is presented in Fig 1I.

## XGBoost and MLP models

The XGBoost and MLP algorithms were used to analyze the data obtained about the adventitious induction of buds and the proliferation of shoots from scale explants of *L. akkusianum*. The outcomes were validated and predicted using three distinct performance metrics. The MLP model yielded results with $R^2$ values ranging from 0.13 to 0.83. Shoot Weight had the lowest $R^2$ ($R^2 = 0.24$), whereas Root Length had the highest ($R^2 = 0.83$). The shoot weight had the lowest Mean Absolute Error (*MAE*) value (0.45), with low values frequently displayed. Root Mean Square Error (*RMSE*) values ranged from 0.60 to 2.74, with shoot weight and shoot length having the lowest and highest *RMSE* values, respectively (Table 3). Regarding the XGBoost model, the root length had the highest $R^2$ ($R^2 = 0.87$), and the shoot weight had the lowest ($R^2 = 0.28$), with $R^2$ values ranging from 0.28 to 0.87. Overall, the *MAE* values ranged from 0.40 to 0.87. The shoot length showed the highest *RMSE* value, and the adventitious bud showed the lowest *RMSE* (Table 4). The distribution of predicted and actual values are presented in Figs 3 and 4. The results suggest that, compared to their relative performances, XGBoost appears to perform better than the MLP model. It demonstrated competitive *RMSE* outcomes, lower *MAE*, greater $R^2$ values, and remarkable performance in predicting root length across all evaluated plant characteristics.

**Table 4. Performance metrices of the MLP and XGBoost models.**

| | Model | RMSE | R² | MAE |
|---|---|---|---|---|
| Adventitious Bud | MLP | 0.61 | 0.74 | 0.51 |
| | XGBoost | 0.52 | 0.84 | 0.42 |
| Root number | MLP | 0.81 | 0.70 | 0.53 |
| | XGBoost | 0.79 | 0.69 | 0.52 |
| Root Length | MLP | 0.73 | 0.83 | 0.51 |
| | XGBoost | 0.65 | 0.87 | 0.44 |
| Shoot number | MLP | 1.33 | 0.18 | 1.07 |
| | XGBoost | 1.05 | 0.32 | 0.87 |
| Shoot length | MLP | 2.74 | 0.35 | 1.81 |
| | XGBoost | 1.21 | 0.80 | 0.88 |
| Shoot weight | MLP | 0.60 | 0.13 | 0.45 |
| | XGBoost | 0.54 | 0.28 | 0.40 |

MLP: Multilayer perceptron; RBF: Radial basis function; XGBOOST: Extreme Gradient Boost; $R^2$: Coefficient of determination; MAE: Mean absolute error; RMSE: Root mean square.

## Discussion

This study focused on the *in vitro* regeneration of *L. akkusianum* and the effects of mT on micropropagation in lilies for the first time in the literature. The highest callus rates were 73.8–83.31% observed in media containing 2.0 mg/L NAA + 0.5 mg/L BA and 2.0 mg/L NAA + 0.5 mg/L mT, respectively. The highest adventitious bud formation per explant was 4.98, obtained from MS medium supplemented with 0.5 mg/L NAA + 1.0 mg/L mT. The results have shown

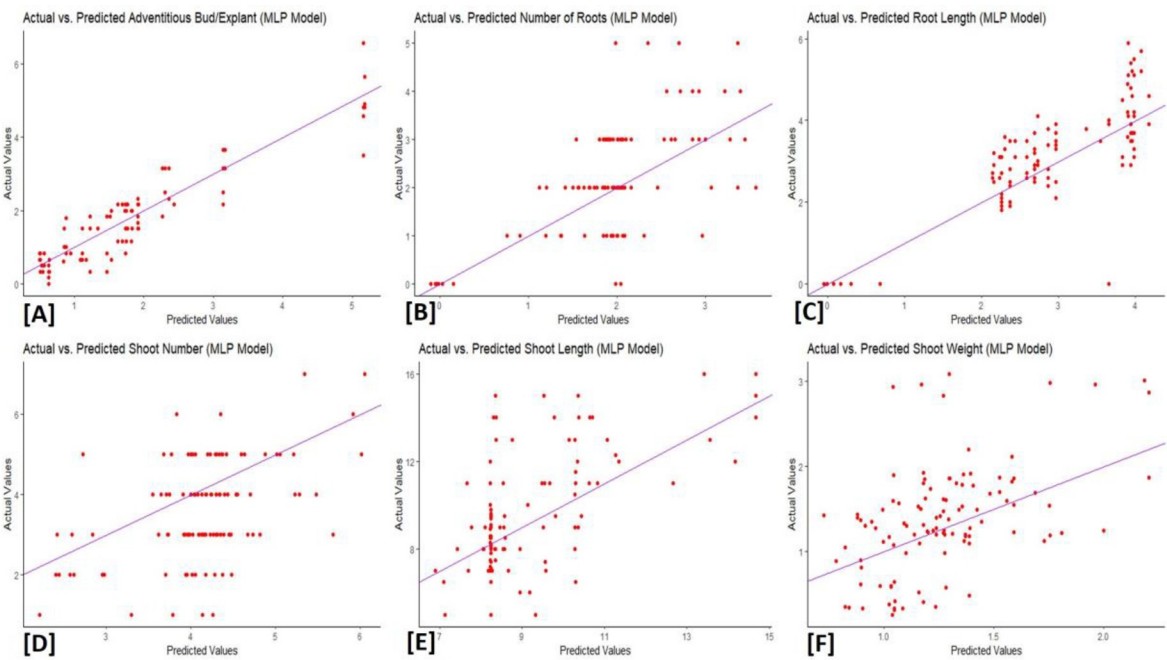

**Fig 3.** Actual against predicted values of (A) adventitious bud, (B) number of roots, (C) root length, (D) number of shoots, (E) shoot length, (F) shoot weight using MLP model analysis.

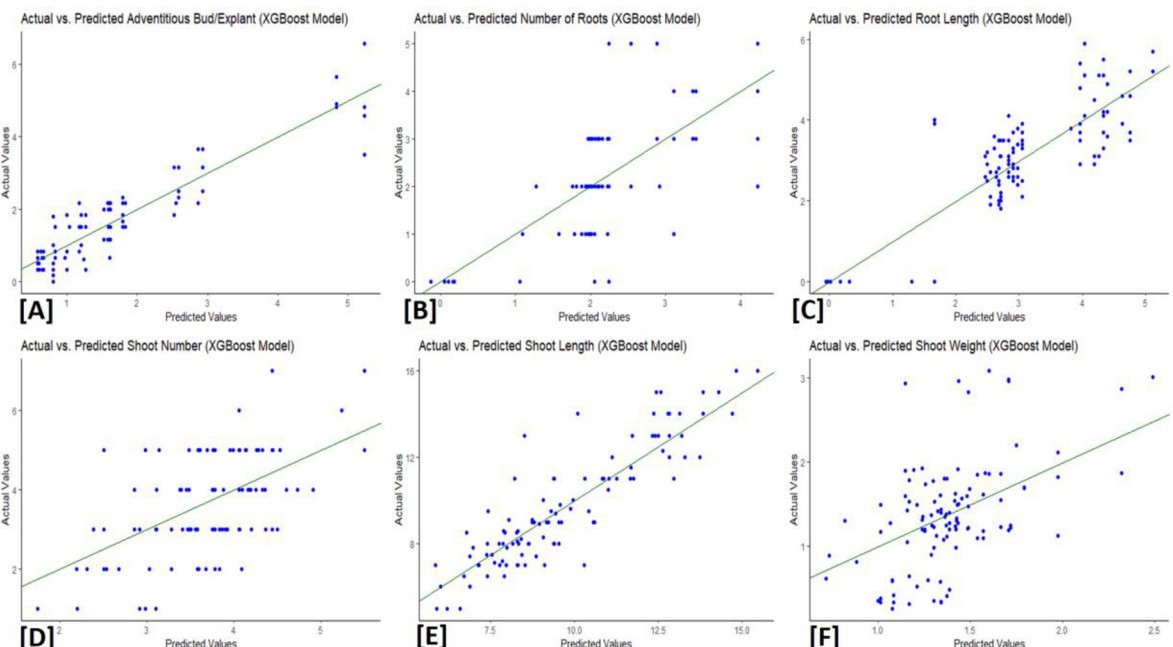

**Fig 4.** Actual against predicted values of (A) adventitious bud, (B) number of roots, (C) root length, (D) number of shoots, (E) shoot length, (F) shoot weight using XGBoost model analysis.

that successful callus and adventitious bud regeneration obtained and MS medium supplemented with NAA and mT instead of BA produced better results.

In previous studies, various types of auxin and cytokinin combinations and concentrations were successfully used in the *in vitro* regeneration of the *Lilium* species. Bakhshaie et al. [41] reported the highest percentage of calluses (65.55%) obtained from scale explants cultured on MS medium supplemented with 0.54 μM NAA and 0.44 μM BA in *L. ledebourii*. Fu et al. [42] observed that MS with 1.0 mg/L NAA and 0.5 mg/L BA were appropriate for inducing callus (64.67%) in *L. brownii* var. *viridulum*. Tang et al. [17] suggested that MS medium containing 0.5 mg/L BA and 1.0–3.0 mg/L 2,4-D was the best for callus induction from scales with the highest percentage (98.3%). Additionally, Nam and Kim [43] indicated that the most successful initiation of adventitious buds from scale explants occurred under dark conditions when placed on an MS medium enriched with 0.5 mg/L BA and 0.1 mg/L NAA. Conversely, the most effective callus induction was observed when microscales were incubated in complete darkness for 8 weeks on MS medium containing 1.0 mg/L 2,4-D and 0.1 mg/L BA. These results showed that the callus induction rate varied mainly based on the medium, species, type, and auxin and cytokinin balance and were in agreement with the current findings in *L. akkusianum* that a higher level of auxin than cytokinin concentration in the medium leads to callus induction. In contrast, higher cytokinin than auxin resulted in adventitious bud induction. Additionally, darkness is a major factor in callus formation because culture initiation in complete darkness can lead to callogenesis due to increased endogenous IAA levels [44]. Therefore, higher cytokinin than auxin in the medium combined with light conditions may be preferred to induce direct regeneration, while higher auxin than cytokinin in the medium and complete darkness for callogenesis and indirect regeneration may be preferred for *L. akkusianum*.

Adventitious buds of *L. akkusianum* were cultured on MS medium supplemented with the combination of NAA, mT and BA. The highest shoot proliferation was obtained from the medium with mT and NAA. However, the effects of the mT or BA combined with NAA in the

medium on shoot weight and shoot length were not statistically significant. Similar studies were conducted in ornamentally valuable bulbous plants to compare the effects of different auxin and cytokinin combinations on *in vitro* shoot proliferation. Sochacki et al. [28] reported that MS medium with 0.1 mg/L 2iP + 0.1 mg/L NAA + 5 mg/L mT produced the highest shoot number with 9.14 per explant in tulip. In another study in *Iris × hollandica* Tub. cv. professor Blaauw, MS + 1.0 mg/L mT + 0.25 mg/L NAA resulted in a considerably higher shoot number per explant with 17.53 and a mean shoot length of 7.06 cm [27]. In agreement with previous studies, MS supplemented with 2.50 mg/L mT with 0.1 mg/L NAA enhanced the formation of healthy and uniformly grown shoots per node and better shoot length in *Withania coagulans* (Stocks) Dunal [45]. In the *in vitro* scale culture of *Cyrtanthus contractus*, a medicinal and ornamental plant, MS medium supplemented with 1–10 μM mT increased the number of shoots per explant, but fresh and dry mass of the shoots decreased in higher concentrations. These results are consistent with the present study for mT.

*In vitro* rooting and acclimatization of *in vitro* grown plantlets are crucial steps for optimizing the regeneration protocol for endemic and under-threat plants. The current study investigated the effects of different concentrations of NAA and IBA on rooting behavior in *L. akkusianum*. Adding IBA into the medium was superior to NAA for rooting rate, root number, and root length. Moreover, rooted plantlets in the medium with IBA have a prolonged survival rate. Superior effects on rooting of *in vitro* grown plantlets in *Lilium* species were also indicated previously. In accordance with the current study, Rafiq et al. [14] compared IBA and NAA performance in rooting of *in vitro* regenerated shoots from scale explants of Oriental *Lilium* Hybrid cv. Ravenna. They reported the highest rooting rate of 92.71%, primary root number per shoot of 12.06, root length of 3.17 cm and survival rate of 98.96% obtained from MS medium supplemented with 1.5 mg/L IBA. Liu and Yang [46] used a half-strength MS medium supplemented with four different IBA concentrations from 0.49 to 4.9 μM for rooting *in vitro* regenerated shoots of *L. orientalis*. They observed that as the concentration of IBA in the medium increased, so did the mean root number, mean root length, and rooting rate. Additionally, medium supplemented with IBA was found to deliver the most favorable results for rooting activity in *L. longiflorum* [21, 47] and *L. napalense* [48]. In contrast, Yang et al. [33] employed IBA and NAA for *in vitro* rooting of shoots in *Lilium davidii* var. *unicolour* Salisb. They discovered that all culture media supplemented with different NAA and IBA resulted in root formation (85–100%). However, NAA was more effective in terms of root number and root length. The results of previous and the present studies may suggest that rooting response to auxin type and concentrations could be related to explants, species and culture conditions. Additionally, a developed root system increases the survival rates of the *in vitro* grown plantlets.

In the present study, the formation of callus tissues was initiated from the epidermal layer, and adventitious buds eventually developed. Callus formation is not desired in organogenesis due to the potential for somatic variations to arise in the shoots regenerated from callus tissue [32]. However, the genetic stability of the regenerated plantlets was assessed using the ISSR marker with 14 primers in *L. akkusianum*; no polymorphic band was detected among the plantlets. The genetic integrity of *in vitro* grown plantlets was previously evaluated using the ISSR marker by Kedra and Bach [49] in *L. martagon* and *L. longiflorum* Oriental hybrid 'Triumphator', *L. davidii* var. *unicolor*, *Lilium* Oriental hybrid 'Siberia', *Lilium* Asiatic hybrid 'Elite' and *L. formolongi* Bi et al. [50]. In agreement with the present study, *in vitro* regenerated plantlets through adventitious bud regeneration from scale explants in lilies can be considered genetically stable.

Cytokinins play a significant role in various physiological aspects of plant tissue culture, including promoting shoot formation, enhancing the quality of explants obtained from *in*

*vitro* culture, influencing plant cell division, differentiation, chlorophyll accumulation, and delaying leaf senescence. Notably, among cytokinins, BA is the most utilized in micropropagation techniques. It is highly effective for *in vitro* shoot propagation, widely employed in commercial laboratories, and more cost-effective than naturally occurring cytokinins [51, 52]. However, using BA to induce shoot production in woody and herbaceous species can lead to challenges such as hyperhydricity, shoot-tip necrosis, inadequate rooting, and reduced survival rates [53, 54]. Moreover, high concentrations of BA have often been implicated in generating genetic diversity among regenerated plantlets [55]. The potential *in vitro* usage of mT as an alternative to BA has been evaluated in many species since its discovery and the initial research published concerning the metabolism and *in vitro* impacts of mT in micropropagated *Spathiphyllum floribundum* [56]. It has been shown that mT improves the multiplication rate, rooting, quality of shoot, and survival ability of in vitro grown plants [57]. In this study, the effects of mT are evaluated for the first time. Different concentrations of mT can be used alone or combined with the low level of auxin for *in vitro* conservation or micropropagation of lilies.

According to our results of ML application, in conjunction with the literature presented by Kirtis et al. [36], Jafari et al. [58], Aasim et al. [59, 60], and Demirel et al. [61], collectively shed light on the diverse applications of machine learning (ML) algorithms in the realm of plant tissue culture and micropropagation. Each study brings unique insights into optimizing growth conditions, predicting outcomes, and enhancing the efficiency of *in vitro* regeneration protocols.

Our study employed XGBoost and MLP algorithms to analyze data related to the adventitious induction of buds and shoot proliferation in *L. akkusianum*. The performance metrics, including $R^2$, Mean Absolute Error (*MAE*), and Root Mean Square Error (*RMSE*), provided a comprehensive evaluation. Notably, XGBoost outperformed MLP regarding competitive *RMSE* outcomes, lower *MAE*, and greater $R^2$ values, particularly excelling in predicting root length across various plant characteristics.

The findings of Kirtis et al. [36] on the *in vitro* whole plant regeneration of desi chickpea underscored the importance of ML algorithms in predicting shoot count and length. The study showcased the effectiveness of Random Forest (RF) in achieving high performance, as indicated by $R^2$ values of 0.99 for shoot count and 0.98 for shoot length. The authors emphasized the utility of RF for precise predictions in tissue culture studies.

Jafari et al. [58] tackled the challenge of *in vitro* adventitious rooting in *Passiflora caerulea*. Integrating a hybrid model, generalized regression neural network (GRNN) with genetic algorithm (GA), successfully predicts rooting responses. The study emphasized the accuracy of the GRNN-GA model, showcasing its reliability in forecasting *in vitro* rooting outcomes. Aasim et al. [60] explored the use of ML algorithms in germinating hemp seeds, highlighting the efficacy of the Random Forest model in predicting output variables with high F1 scores. The study demonstrated the potential of ML in optimizing conditions for seed germination and seedling growth. Similarly, Aasim et al. [59] delved into the *in vitro* regeneration of common beans, combining ML models with artificial neural networks. The Multilayer Perceptron (MLP) model is the most effective in predicting and optimizing output variables, offering valuable insights for biotechnological applications in common bean breeding programs. Demirel et al. [61] explored the optimal conditions for the growth of black chokeberry using tissue culture techniques, employing ML algorithms for predictive modeling. XGBoost and Support Vector Machine (SVM) models stand out in predicting various *in vitro* parameters, showcasing superior performance compared to other models.

These studies underscored the versatility of ML algorithms in predicting, optimizing, and enhancing various aspects of plant tissue culture. The application of ML contributes to the efficiency of in vitro protocols and offers a reliable means of forecasting outcomes, paving the way

for precision agriculture and biotechnological advancements in plant sciences. Integrating ML into plant tissue culture studies holds immense potential for improving the success rates of micropropagation and addressing challenges in the field.

## Conclusions

In conclusion, *L. akkusianum* R. Gämperle, an endemic *Lilium* species, successfully propagated from *in vitro* scale culture through in direct regeneration for the first time. Meta-Topolin, compared to BA, increased adventitious bud formation per explants and shoot proliferation. IBA is more appropriate than NAA in improving rooting, root number, root length, and survival rate of *in vitro* grown plantlets. ISSR markers proved the genetic stability of the plantlets. This protocol is highly efficient inducing callus and shoot regeneration and can potentially conserve the valuable germplasm of endemic *L. akkusianum* genotypes.

This study employed XGBoost and MLP algorithms to model and predict adventitious bud induction, shoot proliferation, and rooting from scale explants of *L. akkusianum*. XGBoost outperformed MLP in terms of competitive *RMSE* outcomes, lower *MAE*, and greater $R^2$ values, which means it is a more suitable model than MLP for micropropagation of *L. akkusianum*. As forthcoming investigations explore the synergy between machine learning (ML) and *in vitro* techniques, there exists the possibility of revealing innovative insights and strategies to enhance the sustainability and efficacy of plant propagation methods. The integration of machine learning into lily micropropagation signifies a forward-thinking approach to leveraging state-of-the-art technologies for the preservation and utilization of genetic resources in agriculture and conservation.

## Supporting information

**S1 Fig. Raw images of DNA bands profile of *in vitro* regenerated plantlets.** Raw images of Fig 1I given. DNA bands profile from left to right; fist is column DNA ladder; second column is mother plant and others belong to *in vitro* regenerated plants.
(TIF)

**S2 Fig. Effects different medium on shoot number, fresh shoot height and shoot weight.** Raw data set belonging to shoot number, fresh shoot height and shoot weight from adventitious buds cultured on MS medium containing 1 mg/L NAA + 2 mg/L mT, 1 mg/L NAA + 2 mg/L BA and 0.5 mg/L NAA + 1 mg/L mT.
(DOCX)

**S3 Fig. Codes for MLP model analysis.**
(TXT)

**S4 Fig. Codes for XGBoost model analysis.**
(TXT)

**S1 Table. Callus induction rate and adventitious bud number per explants.** Raw data set belonging to callus induction rate and adventitious bud number per explants cultured on the MS medium supplemented with various concentrations of NAA and BA or mT.
(DOCX)

**S2 Table. Rooting behavior and survival rate of plantlets.** Raw data set belonging to root number, root length and survival rate of the plantlets.
(DOCX)

## Acknowledgments

The author would like to thank Birol Kurt, Ondokuz Mayıs University, R&D Coordination Office, Academic Writing Advisory Unit, for the English language editing of the manuscript; Dr. Hakan Yilmaz, Department of Forestry, Akkuş Vocational School, Ordu University, for the botanical identification of *L. akkusianum*; Dr. Özhan Şimşek, Horticulture Department, Erciyes University for the technical support in molecular techniques.

## Author Contributions

**Conceptualization:** Mehmet Tütüncü.

**Investigation:** Mehmet Tütüncü.

**Methodology:** Mehmet Tütüncü.

**Writing – original draft:** Mehmet Tütüncü.

**Writing – review & editing:** Mehmet Tütüncü.

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
