## [Decision Letter · Decision Letter 0]

10 Mar 2024

PONE-D-24-00457Integrating machine learning in endemic lily conservation: advanced micropropagation techniques for Lilium akkusianum R. GämperlePLOS ONE

Dear Dr. Tütüncü,

Thank you for submitting your manuscript to PLOS ONE. After careful consideration, we feel that it has merit but does not fully meet PLOS ONE’s publication criteria as it currently stands. Therefore, we invite you to submit a revised version of the manuscript that addresses the points raised during the review process.

We look forward to receiving your revised manuscript.

Kind regards,

Mojtaba Kordrostami, Ph.D.

Academic Editor

PLOS ONE

6. We note that Figure 1 in your submission contain copyrighted images. All PLOS content is published under the Creative Commons Attribution License (CC BY 4.0), which means that the manuscript, images, and Supporting Information files will be freely available online, and any third party is permitted to access, download, copy, distribute, and use these materials in any way, even commercially, with proper attribution. For more information, see our copyright guidelines: http://journals.plos.org/plosone/s/licenses-and-copyright.

Additional Editor Comments:

Thank you for submitting your manuscript titled Integrating machine learning in endemic lily conservation: advanced micropropagation techniques for Lilium akkusianum R. Gämperle to PLOS ONE. We have now completed the review of your manuscript, and after considering the feedback from 2 experts in the field, we believe that your study makes a valuable contribution to the knowledge on the micropropagation of Lilium akkusianum. However, before your manuscript can be considered for publication, it is necessary for you to address several important issues raised during the review process. Therefore, we invite you to resubmit your manuscript after making major revisions.

Reviewer 1 Comments:

Writing Quality: There are concerns about the clarity and quality of writing throughout the manuscript. It is recommended that you seek the assistance of an expert in scientific writing to improve the English language and presentation of your work.

Species Name Formatting: Please ensure that the species name, Lilium akkusianum, is presented in full and italicized upon its first mention in the abstract, and maintain correct formatting throughout the text.

Introduction Section: The introduction should focus specifically on research progress related to Lilium akkusianum rather than a broad overview of tissue culture in different species. Any comparison with other species should be moved to the discussion section.

Histological Sections: The relevance and quality of the histological sections presented are questioned. It is advised to remove these sections, except for Figure 3G, which was deemed valuable.

Formatting: Ensure that all instances of Latin names and terms such as in vitro/ex vitro are italicized throughout the manuscript.

Reviewer 2 Comments:

Title Revision: The current title misleadingly suggests that machine learning (ML) is a micropropagation technique. Since ML is used as an analysis method in your research, the title should be revised to accurately reflect its application in the study of Lilium akkusianum. A suggested revision could focus on the in vitro propagation of Lilium akkusianum and the application of ML in this context.

Conclusions: The conclusions need to be revisited to provide specific insights into the ML methods applied and their effectiveness in the study. General statements on the use of ML should be supplemented with concrete findings from your research.

General Recommendations:

Ensure consistent formatting for scientific names and terms throughout the manuscript.

Review and address the specific points for revision mentioned in the attached PDF by the second reviewer.

We appreciate the effort you have put into your research and believe that addressing these points will significantly enhance the quality and impact of your manuscript. Please submit your revised manuscript along with a detailed response to the reviewers' comments and a summary of the changes made.

We look forward to receiving your revised manuscript and are optimistic about its potential contribution to the field. Should you have any questions or require further clarification, please do not hesitate to contact the editorial office.

Sincerely,

Mojtaba Kordrostami

Editor, PLOS ONE

Reviewers' comments:

Reviewer's Responses to Questions

**Comments to the Author**

1. Is the manuscript technically sound, and do the data support the conclusions?

Reviewer #1: Partly

Reviewer #2: Yes

2. Has the statistical analysis been performed appropriately and rigorously? 

Reviewer #1: N/A

Reviewer #2: Yes

3. Have the authors made all data underlying the findings in their manuscript fully available?

Reviewer #1: Yes

Reviewer #2: Yes

4. Is the manuscript presented in an intelligible fashion and written in standard English?

Reviewer #1: No

Reviewer #2: Yes

5. Review Comments to the Author

Reviewer #1: A primary adventitious buds and regeneration protocol was developed for micropropagation of Lilium akkusianum bulb explants.

They have finished callus induction, bud regeneration and transplantation process. Howerver, the writing has some problem and need improve.

It is hoped that the author will find an expert to revise the writing of the paper and improve the English level；

In the abstract, when the species name appears first, the full name is concerned;

In the introduction, there is no need to introduce the relevant research on the tissue culture of other species, as long as the research progress of that species is introduced, you can transplant them into the discussion to take the discussion, and introduce their similarities and differences；

Histological sections don't make any sense here, and they aren't done well; It is advisable to just keep Figure 3G;

Major latin requires italics, note spaces;

Reviewer #2: Minor Revision

General Comments

The study presents a suitable micropropagation method for Lilium akkusiamum, using various phytohormones.

This is the first study on this endemic species.

The genetic stability of regenerated plantlets was assessed as well by molecular markers.

Machine learning methods were evaluated as well.

The reference list is extended with recent manuscripts.

There are no unnecessary references.

A general comment is that in vitro/ex vitro etc and scientific names must be written in italics everywhere in the text.

Introduction

The background is OK. There are information relevant studies and in vitro methods in the genus Lilium. The aim is clearly presented.

Material and Methods

Well and clear presented.

Statistical methods, design and replications suitable for analysis.

Results

Well presented, including the necessary figures, tables and pictures.

Discussion

Well presented. It contains the necessary references.

There are some issues in the attached .pdf

To my opinion there are two points to be revised:

1. Title

“Integrating machine learning in endemic lily conservation: advanced micropropagation techniques for Lilium akkusianum R. Gämperle”

Comment:

Machine learning is not a micropropagation technique. It is an analysis method used in micropropagation research. The reports confirm the reliability and accuracy of artificial intelligence in micropropagation systems. On the other hand, they are methods of analyzing data. I mean that micropropagation techniques are meristem culture, embryo culture, callus culture e.t.c. Additionally the manuscript presents a valuable, conventional, micropropagation method by scales and various phytohormones on solid MS media. To my opinion, there have not been used advanced micropropagation techniques and the title should be revised. It is about an application of ML in in vitro propagation of endemic lily Lilium akkusianum R. Gämperle.

The title must be focused on in vitro propagation and the usage/application of ML.

2. Conclusions

Comment:

I think that they need revision. There are general comments on the use of ML. Did a suitable ML-method revealed?

6. PLOS authors have the option to publish the peer review history of their article (what does this mean?). If published, this will include your full peer review and any attached files.

Reviewer #1: No

Reviewer #2: No

---

## [Author Response · Author response to Decision Letter 0]

20 May 2024

Response to Editor

1- The revised manuscript considering PLOS ONE's style requirements and reviewer comments uploaded journal system. 

2- In method section, I gave additional information regarding the permits for our plant material as “L. akkusianum bulbs were gathered from the natural habitat in Akkus province, Ordu, Türkiye after the flowering season in 2021 with the official permission numbered 21264211-288.04-E.817744 obtained from the Ministry of Agriculture and Forestry, the Republic of Türkiye”.

3- Author generated codes for machine learning methods provided.

4- Minimal data set were prepared for graphs and tables and uploaded to system as Supporting information files.

5- Original uncropped and unadjusted image underlying gel result in Fig 1 reported in manuscript provided.

6- You have noted that Figure 1 in our submission contain copyrighted images. However, all images including Figure 1 are original photographs taken by me during the laboratory works and I have also revised some of them with the similar photographs. 

The title of the manuscript revised considering comments of reviewer and histological section of the manuscript was removed. The language of the manuscript was edited Academic Writing Advisory Unit of the University.

Response to Reviewers

Reviewer 1

Comment: Writing Quality: There are concerns about the clarity and quality of writing throughout the manuscript. It is recommended that you seek the assistance of an expert in scientific writing to improve the English language and presentation of your work.

Response: Thank you for your point out. The English language of revised manuscript edited by Birol Kurt who is professional language translator at Ondokuz Mayıs University, R&D Coordination Office, Academic Writing Advisory Unit.

Comment: Species Name Formatting: Please ensure that the species name, Lilium akkusianum, is presented in full and italicized upon its first mention in the abstract, and maintain correct formatting throughout the text.

Response: Thank you. All species name checked and revised throughout the text.

Comment: Introduction Section: The introduction should focus specifically on research progress related to Lilium akkusianum rather than a broad overview of tissue culture in different species. Any comparison with other species should be moved to the discussion section.

Response: Thank you for your valuable suggestion, but we don’t agree with you. This is the first report on the propagation of endemic lily species. Also, literature about the usage of meta-topolin in in vitro propagation in ornamental plants is very limited. Therefore, we believe that giving literature rather than L. akkusianum species but close to current studies in introduction part will provide a background to readers to understand why this study was conducted clearly.

Comment: Histological Sections: The relevance and quality of the histological sections presented are questioned. It is advised to remove these sections, except for Figure 3G, which was deemed valuable.

Response: Thank you for your contributions and advice. Histological section removed from the manuscript, and Figure 3G integrated with Figure 1.

Comment: Formatting: Ensure that all instances of Latin names and terms such as in vitro/ex vitro are italicized throughout the manuscript.

Response: Thank you. All Latin names checked and italicized throughout the text.

Comment: Machine learning is not a micropropagation technique. It is an analysis method used in micropropagation research. The reports confirm the reliability and accuracy of artificial intelligence in micropropagation systems. On the other hand, they are methods of analyzing data. I mean that micropropagation techniques are meristem culture, embryo culture, callus culture e.t.c. Additionally the manuscript presents a valuable, conventional, micropropagation method by scales and various phytohormones on solid MS media. To my opinion, there have not been used advanced micropropagation techniques and the title should be revised. It is about an application of ML in in vitro propagation of endemic lily Lilium akkusianum R. Gämperle.

The title must be focused on in vitro propagation and the usage/application of ML.

Comment: Thank you very much for your valuable comments. I have revised the title as “Application of machine learning in in vitro propagation of endemic Lilium akkusianum R. Gämperle”

Reviewer #2: 

Comment: General Comments The study presents a suitable micropropagation method for Lilium akkusiamum, using various phytohormones. This is the first study on this endemic species. The genetic stability of regenerated plantlets was assessed as well by molecular markers. Machine learning methods were evaluated as well. The reference list is extended with recent manuscripts. There are no unnecessary references. A general comment is that in vitro/ex vitro etc and scientific names must be written in italics everywhere in the text.

Response: Thank you very much for your efforts and valuable contributions to study. All scientific names checked and rewritten in italics.

Introduction

Comment: The background is OK. There are information relevant studies and in vitro methods in the genus Lilium. The aim is clearly presented.

Response: Thank you for your comment.

Comment: Material and Methods Well and clear presented. Statistical methods, design and replications suitable for analysis.

Comment: Results: Well presented, including the necessary figures, tables and pictures.

Discussion: Well presented. It contains the necessary references.

There are some issues in the attached .pdf

To my opinion there are two points to be revised:

1. Title

“Integrating machine learning in endemic lily conservation: advanced micropropagation techniques for Lilium akkusianum R. Gämperle”

Comment:

Machine learning is not a micropropagation technique. It is an analysis method used in micropropagation research. The reports confirm the reliability and accuracy of artificial intelligence in micropropagation systems. On the other hand, they are methods of analyzing data. I mean that micropropagation techniques are meristem culture, embryo culture, callus culture e.t.c. Additionally the manuscript presents a valuable, conventional, micropropagation method by scales and various phytohormones on solid MS media. To my opinion, there have not been used advanced micropropagation techniques and the title should be revised. It is about an application of ML in in vitro propagation of endemic lily Lilium akkusianum R. Gämperle.

The title must be focused on in vitro propagation and the usage/application of ML.

Response: Thank your vey much your consideration. I agree that the revision of title, so title revised as “Application of machine learning in in vitro propagation of endemic Lilium akkusianum R. Gämperle”

2. Conclusions

Comment:

I think that they need revision. There are general comments on the use of ML. Did a suitable ML-method revealed?

Response: Thank you very much for your point out this. It is stated in the discussion section which model gives better results. However, it was missing in conclusion part as you stated. Therefore, we stated it in conclusion part as; “In this study, XGBoost and MLP algorithms were employed to model and predict adventitious bud induction, shoot proliferation, and rooting from scale explants of L. akkusianum. XGBoost outperformed MLP in terms of competitive RMSE outcomes, lower MAE, and greater R2 values which means it is a more suitable model than MLP for micropropagation of L. akkusianum.”

---

## [Decision Letter · Decision Letter 1]

27 May 2024

PONE-D-24-00457R1Application of machine learning in in vitro propagation of endemic Lilium akkusianum R. GämperlePLOS ONE

Dear Dr. Tütüncü,

Thank you for submitting your manuscript to PLOS ONE. After careful consideration, we feel that it has merit but does not fully meet PLOS ONE’s publication criteria as it currently stands. Therefore, we invite you to submit a revised version of the manuscript that addresses the points raised during the review process.

We look forward to receiving your revised manuscript.

Kind regards,

Mojtaba Kordrostami, Ph.D.

Academic Editor

PLOS ONE

Journal Requirements:

Additional Editor Comments:

Dear colleagues,

Please revise the manuscript before the final acceptance.

Regards

M. Kordrostami

Reviewers' comments:

Reviewer's Responses to Questions

**Comments to the Author**

1. If the authors have adequately addressed your comments raised in a previous round of review and you feel that this manuscript is now acceptable for publication, you may indicate that here to bypass the “Comments to the Author” section, enter your conflict of interest statement in the “Confidential to Editor” section, and submit your "Accept" recommendation.

Reviewer #1: All comments have been addressed

Reviewer #2: All comments have been addressed

2. Is the manuscript technically sound, and do the data support the conclusions?

Reviewer #1: Yes

Reviewer #2: Yes

3. Has the statistical analysis been performed appropriately and rigorously? 

Reviewer #1: Yes

Reviewer #2: Yes

4. Have the authors made all data underlying the findings in their manuscript fully available?

Reviewer #1: Yes

Reviewer #2: Yes

5. Is the manuscript presented in an intelligible fashion and written in standard English?

Reviewer #1: Yes

Reviewer #2: (No Response)

6. Review Comments to the Author

Reviewer #1: After the author can revised the following mini errors in the text, the manuscript can be accepted.

Line 199 µ?

Line 199, subscript;

The second paragraph or others need to be preceded by a few spaces；

Between two numbers in the text, use “‒”;

Conclusion section, it's too long, reducing the numbers and condensing the content.

Reviewer #2: (No Response)

7. PLOS authors have the option to publish the peer review history of their article (what does this mean?). If published, this will include your full peer review and any attached files.

Reviewer #1: No

Reviewer #2: No

---

## [Author Response · Author response to Decision Letter 1]

24 Jun 2024

Response to Reviewers

Comment: Reviewer #1: After the author can revised the following mini errors in the text, the manuscript can be accepted.

Line 199 µ?

Line 199, subscript;

The second paragraph or others need to be preceded by a few spaces；

Between two numbers in the text, use “‒”;

Conclusion section, it's too long, reducing the numbers and condensing the content.

Response: Thank you very much your valuable contribution. Mini errors in the text corrected. Units given in Line 199 re-checked and both are correct, therefore they were not changed.

We left spaces between paragraphs and used “‒” between two numbers in the text. Conclusion section revised and shortened.

---

## [Editor Report · Decision Letter 2]

12 Jul 2024

Application of machine learning in in vitro propagation of endemic Lilium akkusianum R. Gämperle

PONE-D-24-00457R2

Dear Dr. Tütüncü,

We’re pleased to inform you that your manuscript has been judged scientifically suitable for publication and will be formally accepted for publication once it meets all outstanding technical requirements.

Kind regards,

Mojtaba Kordrostami, Ph.D.

Academic Editor

PLOS ONE